# The Association and Significance of p53 in Gynecologic Cancers: The Potential of Targeted Therapy

**DOI:** 10.3390/ijms20215482

**Published:** 2019-11-04

**Authors:** Mitsuhiro Nakamura, Takeshi Obata, Takiko Daikoku, Hiroshi Fujiwara

**Affiliations:** 1Department of Obstetrics and Gynecology, Kanazawa University Graduate School of Medical Sciences, Kanazawa 920-8641, Japan; t-oba.0919@hotmail.co.jp (T.O.); fuji@kuhp.kyoto-u.ac.jp (H.F.); 2Institute for Experimental Animals, Kanazawa University Advanced Science Research Center, Kanazawa University, Kanazawa 920-8641, Japan; tdaikoku@kiea.m.kanazawa-u.ac.jp

**Keywords:** cervical cancer, HPV, endometrial cancer, p53 overexpression, TP53 mutation, ovarian cancer, p53 signature, STIC, gene therapy

## Abstract

Dysfunction of p53 is observed in the many malignant tumors. In cervical cancer, p53 is inactivated by degradation through the complex with human papilloma virus (HPV) oncoprotein E6 and E6-associated protein (E6AP), an E3 ubiquitin protein ligase. In endometrial cancer, overexpression of p53 in immunohistochemistry is a significant prognostic factor. A discrepancy between p53 overexpression and TP53 mutations is observed in endometrioid endometrial cancer, indicating that the accumulation of p53 protein can be explained by not only gene mutations but also dysregulation of the factors such as ERβ and MDM2. Furthermore, the double-positive expression of immunoreactive estrogen receptor (ER) β and p53 proteins is closely associated with the incidence of metastasis and/or recurrence. High-grade serous ovarian carcinoma (HGSC) arises from secretary cells in the fallopian tube. The secretary cell outgrowth (SCOUT) with TP53 mutations progresses to HGSC via the p53 signature, serous intraepithelial lesion (STIL), and serous intraepithelial carcinoma (STIC), indicating that TP53 mutation is associated with carcinogenesis of HGSC. Clinical application targeting p53 has been approved for some malignant tumors. Gene therapy by the adenovirus-mediated p53 gene transfer system is performed for head and neck cancer. A clinical phase III trial using MDM2/X inhibitors, idasanutlin (RG7388) combined with cytarabine, is being performed involving relapse/refractory acute myeloid leukemia patients. The use of adenoviruses as live vectors which encode wild-type p53 has given promising results in cervical cancer patients.

## 1. Introduction

In 1979, p53 protein was first discovered as a 53-k Dalton protein from SV40 transformed cells [1]. It had been thought that p53 was an oncoprotein; however, p53 was recognized as a tumor suppressor protein in 1992 [2].

p53 plays an important role in the regulating of cell proliferation, DNA repair, apoptosis, genomic stability, senescence, and metabolic homeostasis [3]. p53 protein activated by several signals, such as DNA damage, hypoxia, oncogene expression, ribonucleotide depletion, and osmotic stress, acts mainly as a transcriptional factor. When DNA is damaged, p53 induces the expression of p21. p21 is a cyclin-dependent kinase (CDK) inhibitor that suppresses cyclin-CDK complexes, resulting in cell cycle arrest in the G1 phase. G1 arrest can allow DNA repair before replication at S1 [4,5]. If the cells cannot repair the DNA damage, p53 induces apoptosis by activating apoptosis signal genes, such as BAX, PUMA, Noxa, and PERP [6]. Loss of p53 function allows abnormal cell proliferation and is closely associated with carcinogenesis. Dysfunction of p53 has been observed in many malignant tumors [7,8].

Dysfunction of p53 in malignant tumors is mainly due to the inactivation of p53 protein by binding proteins or TP53 mutations. MDM2 plays the role of a negative regulator by direct binding. Mdm2 protein acts both as an E3 ubiquitin ligase that recognizes the N-terminal trans-activation domain (TAD) of the p53 tumor suppressor and as an inhibitor of p53 transcriptional activation. In response to DNA damage, MDM2 releases p53, resulting in its activation [9]. In some malignant tumors such as retinoblastoma and sarcoma, p53 is inactivated by the amplification of MDM2/X [10]. In cervical cancer, p53 is inactivated by Human papilloma virus (HPV) oncoprotein E6 [11]. Results regarding the prognostic impact of p53 over-expression in granulosa cell tumors are conflicting. On the other hand, abnormalities in p53, may be involved in the pathogenesis of small cell carcinomas of the ovary, hypercalcaemic type [12]. TP53 mutations have been observed in about half of patients with malignant tumors [13,14]. About 75% of TP53 mutations were missense mutations. In gynecologic cancer cell lines, HPV-negative cervical cancer cell line, C33A has missense mutation, c.817C > T in exon8 [15]. Endometrial cancer cell line, AN3CA and ovarian cancer cell line, OVCAR-3 also have missense mutations, c.1165G > T in exon11 and c.743G > A in exon7, respectively [16].

In gynecologic cancers, cervical, endometrial, and ovarian cancers make up more than 95%. In this review, we introduce the significance of p53 and potential clinical applications using p53 in these malignant tumors.

## 2. Significance of p53 in Gynecologic Cancers

### 2.1. Significance of p53 in Cervical Cancer

Cervical cancer occurs due to high-risk HPV infection [17]. Persistent infection by HPV induces the immortalization of cervical cells, appearing pathologically as cervical dysplasia (cervical intraepithelial neoplasia: CIN). Cervical dysplasia progresses to mild (CIN1), moderate (CIN2), and severe (CIN3) dysplasia. A small population of cervical dysplasia progresses to invasive cervical cancer.

HPV is a 6800–8000-base pairs DNA virus that consists of eight open reading frames, E1, E2, E4, E5, E6, E7, L1, and L2. E6 and E7 play important roles in the carcinogenesis of cervical cancer as oncoproteins. When HPV infects epithelial cells in the squamo-columnar junction of the cervix, the E6 and E7 oncogenic protein products act by regulating the cell cycle and apoptosis, resulting in promoting the immortalization of HPV-infected cells [18].

The main function of E7 oncoprotein for immortalization and neoplastic transformation is inactivation of pRB. The complex of dephosphorylated pRB with the transcriptional factor E2F suppresses the cell cycle. The pRB phosphorylated by E7 is released from the complex with E2F, resulting in E2F activation. High activity of E2F induces not only progression of the cell cycle but also apoptosis. E2F stimulates cyclin E. E7 stimulates cyclin-A and E, moving the cells to the S-phase [19]. Cyclin-dependent kinase inhibitors such as p21 and p27 are suppressed in E7-expressing cells [20,21,22]. These regulations promote the cell cycle. E2F upregulates INK4A (known as p16), resulting in the progression of apoptosis in E7-expresing cells [17]. E2F also induces the expression of p14ARF. The p14ARF binds to HDM2 and then this complex represses p53 degradation, resulting in inducing p53-dependent apoptosis [23,24].

The main function of E6 oncoprotein for immortalization and neoplastic transformation is enhancing the ubiquitin-dependent proteosomal degradation of p53 [25]. This degradation occurs through the complex with E6 and E6-associated protein (E6AP), an E3 ubiquitin protein ligase. E6 also directly represses the p53 pathway by binding to transcriptional conjugating factors of p53 such as CBP/p300 and ADA3 [26]. E6 suppresses apoptosis by binding to BAK, FADD, and pro-caspase8 [27,28,29,30,31].

The inactivation of p53 is associated with carcinogenesis of cervical cancer. Notch1 acts as a tumor suppressor gene [32] and induces cellular differentiation in keratinocytes [33]. Notch1 also reduces cell proliferation in cervical cancer cell lines through E6 and E7 oncogenes [34]. Yugata et al. demonstrated that E6 suppressed the expression of Notch1 through inactivation of p53 [35]. p53 directly binds to the Notch1 promoter and regulates the expression of Notch1 at the transcriptional level. In carcinogenesis of cervical cancer, the promotion of p53 degradation by E6 reduces the expression of Notch1. Furthermore, the expression of ErbB2, a proto-oncogene encoding a transmembrane receptor protein, was suppressed by p53 and the accumulation of ErbB2 (known as HER2/neu) through inactivation of p53 by E6 progressed cell proliferation in HPV-infected keratinocytes [36].

E6 and E7 can immortalize cervical cells independently and synergistically (Figure 1). INK4A inhibits the function of E6 whereas E7 suppresses this inhibition. On the other hand, E6 inhibits E7-induced apoptosis by degrading the apoptosis-inducing proteins p53 and BAK 14 [37]. In an in vitro experiment, only E6 or E7-expressing epithelial cells could not be immortalized, supporting the importance of the synchronous function of E6 and E7 in cervical carcinogenesis [38].

Clinically, immunohistochemical analysis of p53 has been performed in cervical lesions [39,40,41,42,43,44,45,46,47]. Silva et al. summarized nine studies on p53 immunohistochemistry in cervical lesions [48]. Four out of the nine studies showed high expression of p53 invasive cervical cancer compared to normal epithelium [40,41,46,47]. Three studies demonstrated that p53 expression in HSIL was significantly higher than that in normal epithelium [40,41,44]. Two studies demonstrated that the positive intensity and cells showing p53 expression were increased in accordance with the progression of CIN [40,44]. However, the remaining studies showed no significant differences among normal cervical epithelium, dysplasia, and cancer [39,42,43,44,45]. Dysfunction of p53 by E6 and E7 leads to the promotion of degradation, not TP53 mutation, so it may be difficult to detect p53 expression by immunohistochemistry.

### 2.2. Significance of p53 in Endometrial Cancer

Endometrial cancer has been classified into type-I and type-II based on clinical, histopathological, and molecular findings [49]. Type-I endometrial cancer mainly consists of endometrioid cancer that is considered to develop in an estrogen-dependent manner [50,51], arises in atypical endometrial hyperplasia, occurs in premenopausal or perimenopausal women, and is associated with a favorable prognosis. Genetically, in type-I endometrial endometrioid carcinoma, dysfunction of DNA mismatch repair genes and gene mutations in PTEN and KRAS have been shown to be associated with carcinogenesis of the endometrium [52,53,54]. In contrast, type-II endometrial cancer mainly consists of serous cancer that is thought to be de novo carcinogenesis developing directly from the atrophic endometrium, occurs in postmenopausal women, and is associated with a poor prognosis [55]. Genetically, TP53 mutation, HER2/neu, and loss of E-Cadherin are more frequent in type-II than type-I endometrial cancer [56].

Dysfunction of p53 in endometrial cancer is closely associated with TP53 mutation. TP53 mutation is detected in about 25% of all endometrial cancer patients [57]. The frequency of TP53 mutation in type I endometrial cancer is about 10–40%, whereas that in a type II endometrial cancer is about 90% [55]. Schulthei et al. analyzed TP53 mutations in a total of 228 cases of endometrial carcinomas, including 186 cases of endometrioid carcinomas and 42 cases of serous carcinomas [58]. TP53 mutations were detected in 64 cases (28%) of endometrial carcinomas. In the total of 186 endometrioid endometrial carcinomas and 42 serous endometrial carcinomas, TP53 mutation was noted in 27 (15%) and 37 (88%) cases, respectively. In 27 endometrioid endometrial carcinomas cases with TP 53 mutations, the pattern of TP53 mutations was: 3 frameshift, 20 missense, and 4 nonsense mutations. In 37 endometrioid endometrial carcinomas cases with TP53 mutations, the pattern of TP53 mutations was: 4 frameshift, 29 missense, 3 nonsense, and 1 splice-site mutation. The most frequent pattern of TP53 mutations was missense mutations, the rate of which was about 75% in both histological types. TP53 mutations in serous endometrial carcinomas were significantly correlated with poor survival, whereas those in endometrioid endometrial carcinomas were not significantly correlated with the outcome in terms of overall survival [58].

The missense mutation of TP53 results in nuclear accumulation of p53 protein that is observed as overexpression in immunohistochemistry. Immunohistochemistry of p53 protein has been performed in many studies. The p53-positive rate is 17–45% in all histological types of endometrial carcinomas. In type-I endometrial cancer, the p53-positive rate was 10–44%, whereas a high rate of p53 expression was observed in 30–86% of type-II endometrial cancer [59,60,61,62,63,64,65,66,67,68,69,70,71,72,73,74,75] (Table 1). Overexpression of p53 is a poor prognostic factor in both type-I and type-II endometrial cancer. When the pattern of TP53 mutations is frameshift and nonsense mutations, p53 immunoreactivity is completely lacking, resulting in the null phenotype of immunoreactive p53 expression.

A discrepancy between immunoreactive expression of p53 and TP53 mutations was reported in several studies including ours. It has been reported that p53 immunoreactive expression, not a normal staining pattern, was observed in endometrioid endometrial carcinomas without TP53 mutations. Stewart et al. demonstrated that 18 endometrioid endometrial carcinoma patients showed diffuse or focal p53 expression in immunohistochemistry, whereas direct sequencing for exon 5–10 of TP53 detected no mutations in any of the 18 patients [76]. Soong et al. showed that the rate of overexpression of p53 in the nucleus was 27% and that in the cytoplasm was 54% in 122 endometrial carcinomas, respectively, whereas TP53 mutation was detected in only 13% [63]. Furthermore, overexpression of p53 was significantly correlated with a poor prognosis and independent prognostic factor in multivariate analysis adjusted for the surgical stage, histological grade and type, and vascular invasion, whereas statistical analysis could not detect an association between TP53 mutation and poorer survival. We performed immunohistochemical analysis of 154 endometrioid endometrial carcinoma patients [75]. The detection of immunoreactive p53 (positive rates more than 10%, p53-stained) was independently associated with metastasis and/or recurrence. We classified the p53-stained group into a low-positive group (10–49%) and high-positive group (50–100%) (Figure 2). The 33 (21%) patients showed 10–49% p53-positive cells in immunohistochemistry. TP53 mutation analysis was performed in 25 of 33 patients. Interestingly, TP53 mutations were detected in only 4 (16%) of 25 patients. Furthermore, only 1 (25%) of 4 patients had metastasis, indicating that the accumulation of p53 protein in endometrioid endometrial cancer can be explained by not only gene mutations but also the dysregulation of factors such as MDM2 or MDMX [77]. Further investigation of these associated factors is necessary to understand the clinical significance of the positive expression of p53 protein in patients with endometrioid carcinoma and a poor prognosis.

Estrogen is a female sex hormone that regulates the endometrium in the uterus with progesterone. The activity of estrogen on the endometrium is mediated through two main estrogen receptor (ER) isoforms, ERα and ERβ. ERα has been reported to play an important role in the development of endometrioid endometrial cancer. Recent studies proposed that ERβ is also associated with gynecologic malignant tumors [78,79]. We demonstrated that high expressions of not only p53 but also ERβ (high-ERβ) in immunohistochemistry were independently associated with metastasis and/or recurrence of disease in endometrioid endometrial cancer [75]. Furthermore, disease-free survival in patients with p53 expression and high-ERβ expression in immunohistochemistry was significantly shorter than that in other patient groups. However, the relationship between p53 and ERβ at the molecular level is not clear. Induction of ERβ expression did not change the p53 expression of p53 wild-type MCF10A cells in response to DNA damage [80]. On the other hand, TP53 mutation was strongly induced in p53-mutated colon cancer SW480 cells, whereas wild-type p53 was strongly downregulated in p53 wild-type colon cancer HCT116 cells in response to ERβ expression [81]. Further clarification of the molecular relationships between ERβ and p53 could provide a rationale for the finding that the double-positive expression of immunoreactive ERβ and p53 proteins is closely associated with the incidence of regional lymph node metastasis and/or postoperative recurrence.

### 2.3. Significance of p53 in Ovarian Cancer

In ovarian malignant tumors, about 90% of the tumors are ovarian epithelial carcinomas. Ovarian epithelial carcinoma has been classified into type-I and type-II according to clinical, histopathological, and molecular findings [82]. Among them, a major histological subtype of type II ovarian epithelial carcinoma is high-grade serous ovarian carcinoma (HGSC). In HGSC, a high frequency of TP53 mutations and immunoreactive expression of p53 were observed [83,84]. HGSC had been thought to arise from ovarian epithelial cells. Recently, it was reported that HGSC originated from the fallopian tube [85].

The change of this concept was due to the identification of BRCA1 and 2 DNA mismatch repair genes [86,87]. About 5–10% of ovarian cancer included hereditary ovarian cancer and about 95% of these patients had BRCA1/2 mutations [88]. BRCA1/2 mutations have a lifetime risk of causing ovarian cancer in 15–65% of carriers at the age of 70 years [89]. Risk-reducing salpingo-oophorectomy (RRSO) has been recommended in women who are BRCA1/2 mutation carriers. In the specimens obtained by RRSO, microscopic HGSC cells were detected in the fallopian tube, named serous intraepithelial carcinoma (STIC) [90]. Several studies described the same TP53 mutations status between STICs and serous carcinomas [91,92,93]. Kindelberger et al. reported that STIC cells were identified in 29 (71%) of 41 serous tubal, peritoneal, and ovarian carcinomas with BRCA1/2 mutations [91]. Ninety-three percent of STICs involved the fimbriae. All 5 cases analyzed showed the same TP53 mutations between STICs and carcinomas [91]. These findings indicate that STIC may be a precursor lesion of pelvic serous carcinoma, including tubal, ovarian, and peritoneal carcinomas, and the carcinogenesis of these carcinomas might be associated with TP53 mutations.

Although STIC has been recognized as a precursor lesion of serous carcinoma, it has been reported that a single-cell epithelial layer with strong p53 immunoreactivity, named the p53 signature, was identified in benign fimbriae regardless of BRCA mutations [92]. The p53 signature was observed predominantly at the fimbriated end of the fallopian tube and targeted the non-ciliated (secretory) cell phenotype. The p53 signature was also frequently associated with STIC [92]. The STIC and p53 signature shared the same TP53 mutations. Immunohistochemical analysis revealed that γ-H2AX staining, expressed by double-strand DNA breakage, was observed in both STIC and p53 signature lesions whereas positive cells of MIB1 and cyclin-E were more frequently present in STIC than the p53 signature [92]. These findings indicate that the p53 signature was initiated by DNA damage such as ionizing radiation and oxidants. In the context of DNA damage and repair in carcinogenesis, the co-localization of the γ-H2AX and p53 immunostaining was observed in the p53 signature. DNA damage activated the ATM/ATR-regulated pathway, which induced cell cycle arrest, and the p53 signature demonstrated a low proliferative activity. However, mutations such as those leading to inactivation of the ATM-Chk2-p53-BRCA1 promoted a high proliferative activity by the expression of MIB1 and cellular atypia, resulting in progression to STIC [92]. Interestingly, there were morphologically intermediate lesions between p53 and STIC, named serous intraepithelial lesions (STIL) [92,94]. Based on these findings, the p53 signature may be a precursor of STIL.

It has been reported that secretary cells in the fallopian tube might be a precursor of the p53 signature, named secretary cell outgrowth (SCOUT) [95]. A SCOUT is a discrete expansion of at least 30 epithelial secretary cells that are not a heterogeneous background of tubal secretory and ciliated cells. In SCOUTs, the frequency of PAX 2 expression was higher than that of p53 expression (96% versus. 25%. respectively) [95]. On the other hand, all cases of p53 signature showed p53 expression. The p53 immunoreactive expression reflected TP53 missense mutations. The SCOUTs with PAX2 and p53 expression were more frequently observed in the fimbria than proximal tube and were contiguous with serous carcinoma. These findings indicate that SCOUTs may be a precursor of the p53 signature and PAX2 may be associated with abnormal SCOUTs. SCOUTs are also associated with age. The number of SCOUT cells in the fallopian tube increased significantly according to increasing age [96]. Age is a significant risk factor of serous carcinoma, indicating that SCOUT may be a potential biomarker for early serous carcinogenesis in the fallopian tube. Figure 3 shows carcinogenesis of ovarian serous carcinoma based on crum’s hypothesis [97].

On the other hand, Pothuri et al. reported that p53 mutational spectrum of epithelial cells in dysplastic lesions like inclusion cysts and deep surface invaginations are seen in the contiguous surface epithelium as well as underlying malignant cells in the ovarian tissues with BRCA heterozygotes. Further analysis may provide a mechanism of carcinogenesis in HGSC [98].

Loss of 53BP1 function by either mutation or downregulation provides PARP inhibitor resistance. Inactivation of downstream factors of 53BP1-mediated repair, also leads to the restoration of DNA end resection, and consequently promotes homology-mediated repair. There is preclinical evidence that the loss of 53BP1 function allows for the partial restoration of homologous recombination in BRCA1-deficient cells and counteracts sensitivity to the PARP inhibitor [99].

PIK3CA gene amplification has also been associated with p53 mutation. It has been demonstrated from ovarian cancer cell lines that activation of the PI3K/AKT pathway may lead to chemotherapy resistance. OLAPCO study targets homology-directed DNA repair defects based on tumor profiling results. Among the 4 treatment arms the combination of olaparib was included with the wee1 inhibitor AZD1775 for those with TP53 mutations [100].

## 3. Clinical Application Using p53

TP53 mutations have been observed in about half of malignant tumors [13,14]. Overexpression of wild-type p53 may induce apoptosis in malignant cells. We introduce gene therapies targeting p53 in malignant tumors.

The first report of gene therapy using p53 was wild-type p53 gene transfer by retrovirus vectors to tumors of patients with lung cancer [101]. Then, the adenovirus-mediated p53 gene transfer system was developed. Adenovirus can replicate in cells whereas retrovirus cannot. Gene transfer by adenovirus had been more effective than that by retrovirus. A phase I clinical trial of repeated intratumoral delivery of adenoviral p53 was performed in patients with advanced non-small cell lung cancer [102].

Non-replicative p53 gene transfer adenovirus vector (Ad5CMV-p53, Advexin) was developed [103] because severe side effects caused by replication of adenovirus vector in the cells were reported. Two phase III trials of head and neck cancer using Advexin were performed at about 80 sites in the US, Canada, and Europe [104]. The China Food and Drug Administration (CFDA) approved Gendicine (adenoviral serotype 5 mediated delivery of a human P53 gene) in 2003 for head and neck cancer, the first gene therapy [105]. So far, Gendicine has been used commercially in more than 30,000 patients [106].

Recently, Yamasaki et al. developed OB-P702 [107], a telomerase-specific replication-competent adenovirus with expression of wild-type p53. Telomerase is activated in many malignant tumors, including gynecologic carcinomas [108,109]. OBP-702 can replicate in cells with telomerase activity. OPB-702 induced p21 suppression by E1A-mediated miR-93/106b upregulation, leading to p53-mediated apoptosis and autophagy in osteosarcoma cells [110].

In some malignant tumors such as retinoblastoma, sarcoma, wild-type p53 is inactivated by overexpression of MDM2 and MDMX [10]. Overexpressed MDM2 and MDMX bind to p53 directly, resulting in dysfunction of p53. In these type of tumors, inhibition of the binding with p53 and MDM2/X is effective for gene therapy. Regarding MDM2/X inhibitors, a clinical phase III trial using combination with idasanutlin (RG7388) and cytarabine, which is a chemotherapeutic agent, is being performed in relapse/refractory acute myeloid leukemia patients [111]. Clinical phase I trials using idasanutlin are going in solid tumors [112]. Idasanutlin is a second-generation MDM2 inhibitor. It belongs to the Nutlins, a group of small molecules, and binds to MDM2 in the p53-binding pocket, resulting in activation of p53 pathways such as cell cycle arrest, and apoptosis and growth inhibition in cancer cells [113].

Adenovirus-mediated p53 gene transfer system may have the potential to be applied in gynecologic malignant tumors. The use of adenoviruses as live vectors which encode wild-type p53 has given promising results. Su et al. reported that the recombinant adenovirus-p53 transfer combined with radiotherapy improved radiotherapeutic survival rates in patients with cervical cancer [114].

Dysfunction of p53 by HPV oncoprotein E6 causes cervical cancer, so it is necessary to develop an E6 inhibitor. Kajitani et al. reported that treatment of HeLa cells with siRNA for HPV E6 permitted adenovirus-mediated transduction of a p53 gene [115]. Dysfunction of pRb by HPV oncoprotein E7 also causes cervical cancer. Recently, it is reported that a member of the pRB family, RBL2 (Retinoblastoma like protein 2 or o130) is associated with cell proliferation. Nor Rashid et al. demonstrated that p130 was associated with cell cycle by binding E7 in cervical cancer cell lines [116]. Liu et al. demonstrated that miR-106—mediated downregulation of RBL2 regulated cellular proliferation and differentiation in High-Grade Serous Ovarian Carcinoma [117]. These researches may lead to new gene therapy targeting p53 in gynecologic malignant tumors.

In ovarian cancer, clinical phase II trials using p53 Synthetic Long Peptides Vaccine with Cyclophosphamide are going [112].

## 4. Conclusions

We reported the relationship between p53 and gynecologic cancers. The dysfunction of p53 is associated with carcinogenesis in cervical and ovarian cancers. In endometrial cancer, it is associated with a poor prognosis, and the accumulation of p53 protein may be associated with not only gene mutations but also dysregulation of factors such as ERβ and MDM2. Furthermore, the double-positive expression of immunoreactive ERβ and p53 proteins is closely associated with the incidence of metastasis and/or recurrence. Regulation of p53 is critical for the treatment of malignant tumors. Gene therapy using p53 has been applied to head and neck cancer. This therapy has the potential for clinical application to treat patients with gynecologic cancers. Further research on gene therapy using p53 will hopefully lead to clinical application for malignant tumors including gynecologic tumors.

It is necessary for the development of molecularly targeted drugs and immune system modulation for patients with persistent, metastatic, and recurrent cervical cancer. Additional biomarkers may help guide therapy for patients who progress on antiangiogenesis therapy in the absence of therapeutic choices. Feldmana et al. demonstrated next-generation sequencing of 224 tumor samples identified mutational hot spots corresponding to PI3KCA (26%), BRCA2 (21%), BRCA1 (10%), KRAS (10%), TP53 (10%), and FBXW7 (10%). Gene amplification of EGFR (11%) and HER2 (8%) was also demonstrated. These results suggest that poly ADP ribose polymerase (PARP) inhibition, PI3K/AKT/mTOR pathway inhibitors, EGFR- and HER2-directed therapy, may be promising areas for future research in advanced cervical cancer [118].

## Figures and Tables

**Figure 1 ijms-20-05482-f001:**
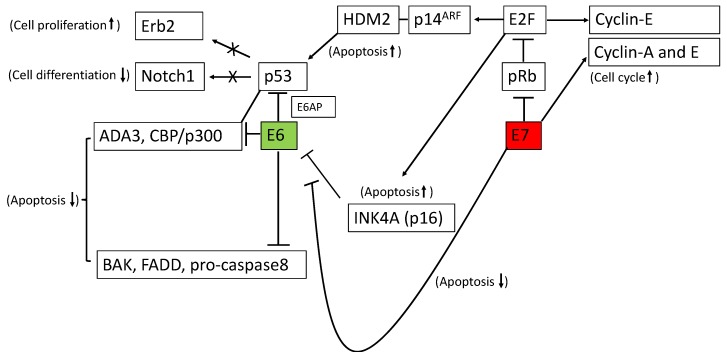
Synergistic functions of E6 and E7 oncoproteins. E6 induces the ubiquitin-dependent proteosomal degradation of p53 by forming a complex with E6 and E6-associated protein (E6AP), resulting in the inhibition of Notch1 and Erb2 expressions. E6 also inhibits transcriptional conjugating factors of p53 such as CBP/p300 and ADA3, and apoptosis-inducing factors such as BAK, far-associated protein with death domain (FADD), and pro-caspase8. E7 inhibits RB, which releases E2F. E2F induces the expression of cyclin-A and E, resulting in cell proliferation. E2F also induces the expression of p14ARF. The complex with p14ARF and human double minute2 (HDM2) represses the p53 degradation, resulting in inducing p53-dependent apoptosis. E7 also inactivates INK4A and rescues E6 from inhibition by INK4A. Up and down arrows, up and down regulation, respectively.

**Figure 2 ijms-20-05482-f002:**
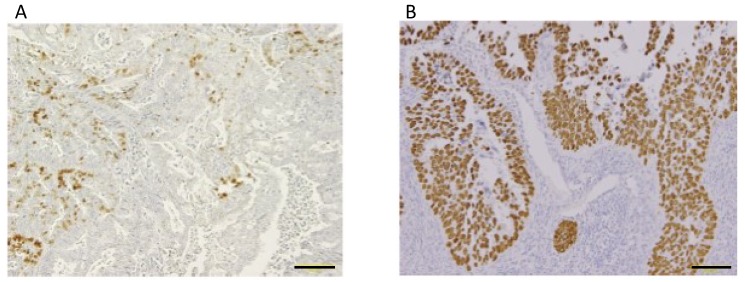
Immunohistochemical localization of p53 proteins in representative cases of endometrial endometrioid carcinoma. **A**, Grade 1 endometrial endometrioid carcinoma with low-positive p53-staining. **B**, Grade 2 endometrial endometrioid carcinoma with high-positive p53-staining. Scale bars: 100 μm.

**Figure 3 ijms-20-05482-f003:**
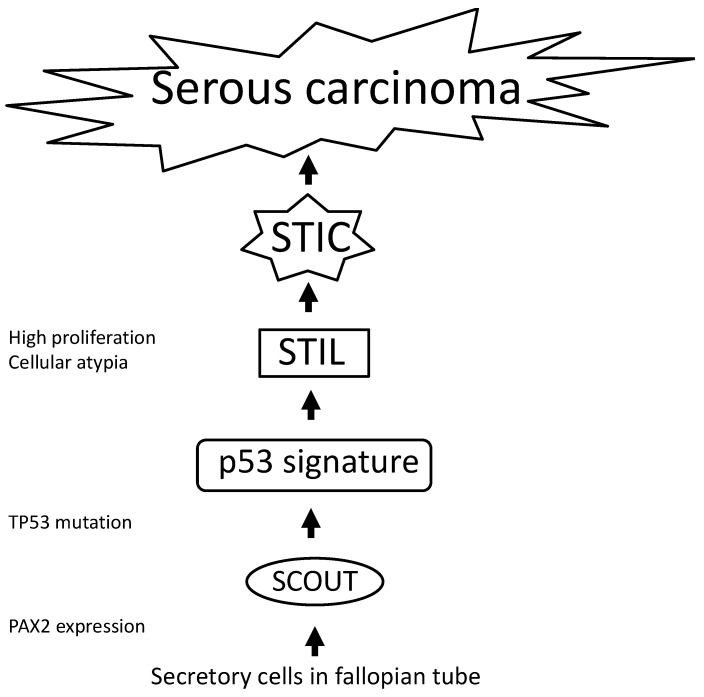
Carcinogenesis of ovarian serous carcinoma. Secretory cell outgrowth (SCOUT) occurs from secretory cells in the fallopian tube, expressing PAX2. The p53 signature develops from SCOUT with TP53 mutation. The p53 signature acquires the activity of high proliferation and cellular atypia, resulting in development of serous tubal intraepithelial carcinoma (STIC) via serous tubal intraepithelial lesions (STILs). STIC progresses to ovarian serous carcinoma.

**Table 1 ijms-20-05482-t001:** The rate of p53 immunoexpression and prognostic impact in endometrial cancer patients.

Authors	Histology	*n*	Positive Rate (%)	Prognostic Factor	Ref
Kohler et al.	End	75	13	N/A	[59]
	Non-End	32	38		
	Total	107	21		
Inoue et al.	End	126	11	Yes	[60]
	Non-End	13	67		
	Total	139	17		
Sherman et al.	End	45	20	N/A	[61]
	Non-End	46	83		
	Total	91	52		
Kohler et al.	End	115	30	Yes	[62]
	Non-End	64	44		
	Total	179	36		
Soong et al.	End	94	19	Yes	[63]
	Non-End	28	54		
	Total	122	27		
Strang et al.	All	183	45	Yes	[64]
Bamcher-Todesca et al.	Non-End	23	48	Yes	[65]
Kouneils et al.	End	40	35	N/A	[66]
	Non-End	21	76		
	Total	61	49		
Coronado et al.	End	87	10	Yes	[67]
	Non-End	27	30		
	Total	114	18		
Shih et al.	All	82	45	Yes	[68]
Suzki et al.	End	112	44	Yes	[69]
Jeon et al.	End	147	20	N.S	[70]
	Non-End	5	40		
	Total	152	20		
Dupont et al.	End	99	14	N/A	[71]
	Non-End	31	41		
	Total	120	21		
Pansare et al.	End	108	17	Yes	[72]
	Non-End	41	82		
	Total	149	35		
Urabe et al.	End	332	17	Yes	[73]
Edmondson et al.	End	86	28	Yes	[74]
	Non-End	28	86		
	Total	114	43		
Obata et al.	End	154	34	Yes	[75]
End; Endometrioid

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
