# Peer review of "The Association and Significance of p53 in Gynecologic Cancers: The Potential of Targeted Therapy"

_ijms, 2019, doi:10.3390/ijms20215482_

Round 1
Reviewer 1 Report
In this manuscript, Nakamura et al showed that the relationship between p53 and gynecologic cancers. These findings are potentially interesting. The manuscript could be further strengthened with a few additional experiments denoted below.
The authors need to explain more about p53 mutation and MDM2 in introduction part. For example, which site is mutated in which cell lines? what is the main function of MDM2? In figure 1, authors need to highlight E6 and E7 otherwise it will be hard to distinguish what is important. Also, please more detail to write into the square such as cyclin-> Cyclin D1 or Cyclin E etc. There are several places that incorrectly or inaccurately write down the manuscript (Please check the space between the words in Conclusions part).
Author Response
Dear reviewer 1
Thank you for your thoughtful and constructive comments. According to your suggestions, we extensively revised our manuscript as follows.
・The authors need to explain more about p53 mutation and MDM2 in introduction part. For example, which site is mutated in which cell lines? what is the main function of MDM2?
We added “In gynecologic cancer cell lines, HPV-negative cervical cancer cell line, C33A has missense mutation, c.817C>T in exon8 [15]. Endometrial cancer cell line, AN3CA and ovarian cancer cell line, OVCAR-3 also have missense mutations, c.1165G>T in exon11 and c.743G>A in exon7, respectively [16].” in introduction part, Line 56 and Ref 15, 16 in revised manuscript.
We also added “Mdm2 protein acts both as an E3 ubiquitin ligase that recognizes the N-terminal trans-activation domain (TAD) of the p53 tumor suppressor and as an inhibitor of p53 transcriptional activation. In response to DNA damage, MDM2 releases p53, resulting in its activation [9].” in introduction part, Line 48 and Ref 9 in revised manuscript.
・In figure 1, authors need to highlight E6 and E7 otherwise it will be hard to distinguish what is important. Also, please more detail to write into the square such as cyclin-> Cyclin D1 or Cyclin E etc.
We highlighted E6 and E7 and wrote “Cyclin A and E” in Figure 1.
・There are several places that incorrectly or inaccurately write down the manuscript (Please check the space between the words in Conclusions part).
We rewrote down the manuscript correctly, spaces etc.
Thank you again for your kindly review and comments.
Reviewer 2 Report
This review highlights the significance of p53 and potential clinical applications using p53 in gynaecological tumors. It is straightforward, well written, concise and updated. Definitely deserves to be published and is a valuable contribution to the “International Journal of Molecular Sciences”. Some minor flaws need to be addressed before publication.
Minor points:
[1] Introduction:
Please, make a comment about the subset of non-epithelial ovarian cancers. Results regarding the prognostic impact of p53 over-expression in granulosa cell tumors are conflicting. On the other hand, abnormalities in p53, may be involved in the pathogenesis of small cell carcinomas of the ovary, hypercalcaemic type.
Recommended reference:
Boussios S, et al. Ovarian sex-cord stromal tumours and small cell tumours: Pathological, genetic and management aspects. Crit Rev Oncol Hematol. 2017 Dec;120:43-51.
[2.3] Significance of p53 in ovarian cancer:
Please, make a comment about the p53 in correlation with BRCA status. Loss of 53BP1 function by either mutation or downregulation provides PARP inhibitor resistance. Inactivation of downstream factors of 53BP1-mediated repair, also leads to the restoration of DNA end resection, and consequently promotes homology-mediated repair. There is preclinical evidence that the loss of 53BP1 function allows for the partial restoration of homologous recombination in BRCA1-deficient cells and counteracts sensitivity to the PARP inhibitor.
Recommended reference:
Boussios S, et al. PARP Inhibitors in Ovarian Cancer: The Route to "Ithaca". Diagnostics (Basel). 2019 May 18;9(2). pii: E55.
[2.3] Significance of p53 in ovarian cancer:
PIK3CA gene amplification has also been associated with p53 mutation. It has been demonstrated from ovarian cancer cell lines that activation of the PI3K/AKT pathway may lead to chemotherapy resistance. OLAPCO study targets homology-directed DNA repair defects based on tumor profiling results. Among the 4 treatment arms was included the combination of olaparib with the wee1 inhibitor AZD1775 for those with TP53 mutations.
Recommended reference:
Boussios S, et al. Combined Strategies with Poly (ADP-Ribose) Polymerase (PARP) Inhibitors for the Treatment of Ovarian Cancer: A Literature Review. Diagnostics (Basel). 2019 Aug 1;9(3). pii: E87.
[3] Clinical application using p53:
The use of adenoviruses as live vectors which encode wild-type p53 has given promising results in vitro.
Recommended reference:
Su X, et al. Effect and Safety of Recombinant Adenovirus-p53 Transfer Combined with Radiotherapy on Long-Term Survival of Locally Advanced Cervical Cancer. Hum Gene Ther. 2016 Dec;27(12):1008-1014.
[4] Conclusions:
Please, make a comment about the unmet need for the development of molecularly targeted drugs and immune system modulation for patients with persistent, metastatic and recurrent cervical cancer. Additional biomarkers may help guide therapy for patients who progress on antiangiogenesis therapy in the absence of therapeutic choices.
In a study, next-generation sequencing of 224 tumour samples identified mutational hot spots corresponding to PI3KCA (26%), BRCA2 (21%), BRCA1 (10%),KRAS (10%), TP53 (10%), and FBXW7 (10%). Gene amplification of EGFR (11%) and HER2 (8%) was also demonstrated. These results suggest that poly ADP ribose polymerase (PARP) inhibition,PI3K/AKT/mTOR pathway inhibitors, EGFR- and HER2-directed therapy, may be promising areas for future research in advanced cervical cancer.
Recommended reference:
Feldman R, et al. Paving the road to personalized medicine in cervical cancer: Theranostic biomarker evaluation in a 592-specimen library. Gynecologic oncology. 2015;137:141. Supplement 1(0).
Author Response
Dear reviewer 2
Thank you for your thoughtful and constructive comments. According to your suggestions, we extensively revised our manuscript as follows.
[1] Introduction:
Please, make a comment about the subset of non-epithelial ovarian cancers. Results regarding the prognostic impact of p53 over-expression in granulosa cell tumors are conflicting. On the other hand, abnormalities in p53, may be involved in the pathogenesis of small cell carcinomas of the ovary, hypercalcaemic type.
Recommended reference:
Boussios S, et al. Ovarian sex-cord stromal tumours and small cell tumours: Pathological, genetic and management aspects. Crit Rev Oncol Hematol. 2017 Dec;120:43-51.
We added “Results regarding the prognostic impact of p53 over-expression in granulosa cell tumors are conflicting. On the other hand, abnormalities in p53, may be involved in the pathogenesis of small cell carcinomas of the ovary, hypercalcaemic type.” in introduction part, Line 52 and above reference (Ref 12) in revised manuscript.
[2.3] Significance of p53 in ovarian cancer:
Please, make a comment about the p53 in correlation with BRCA status. Loss of 53BP1 function by either mutation or downregulation provides PARP inhibitor resistance. Inactivation of downstream factors of 53BP1-mediated repair, also leads to the restoration of DNA end resection, and consequently promotes homology-mediated repair. There is preclinical evidence that the loss of 53BP1 function allows for the partial restoration of homologous recombination in BRCA1-deficient cells and counteracts sensitivity to the PARP inhibitor.
Recommended reference:
Boussios S, et al. PARP Inhibitors in Ovarian Cancer: The Route to "Ithaca". Diagnostics (Basel). 2019 May 18;9(2). pii: E55.
W added “Loss of 53BP1 function by either mutation or downregulation provides PARP inhibitor resistance. Inactivation of downstream factors of 53BP1-mediated repair, also leads to the restoration of DNA end resection, and consequently promotes homology-mediated repair. There is preclinical evidence that the loss of 53BP1 function allows for the partial restoration of homologous recombination in BRCA1-deficient cells and counteracts sensitivity to the PARP inhibitor.” in Significance of p53 in ovarian cancer part, Line 261 and above reference (Ref 99) in revised manuscript.
[2.3] Significance of p53 in ovarian cancer:
PIK3CA gene amplification has also been associated with p53 mutation. It has been demonstrated from ovarian cancer cell lines that activation of the PI3K/AKT pathway may lead to chemotherapy resistance. OLAPCO study targets homology-directed DNA repair defects based on tumor profiling results. Among the 4 treatment arms was included the combination of olaparib with the wee1 inhibitor AZD1775 for those with TP53 mutations.
Recommended reference:
Boussios S, et al. Combined Strategies with Poly (ADP-Ribose) Polymerase (PARP) Inhibitors for the Treatment of Ovarian Cancer: A Literature Review. Diagnostics (Basel). 2019 Aug 1;9(3). pii: E87.
We added “PIK3CA gene amplification has also been associated with p53 mutation. It has been demonstrated from ovarian cancer cell lines that activation of the PI3K/AKT pathway may lead to chemotherapy resistance. OLAPCO study targets homology-directed DNA repair defects based on tumor profiling results. Among the 4 treatment arms was included the combination of olaparib with the wee1 inhibitor AZD1775 for those with TP53 mutations in Significance of p53 in ovarian cancer part, Line 266 and above reference (Ref 100) in revised manuscript.
[3] Clinical application using p53:
The use of adenoviruses as live vectors which encode wild-type p53 has given promising results in vitro.
Recommended reference:
Su X, et al. Effect and Safety of Recombinant Adenovirus-p53 Transfer Combined with Radiotherapy on Long-Term Survival of Locally Advanced Cervical Cancer. Hum Gene Ther. 2016 Dec;27(12):1008-1014.
We added “Adenovirus-mediated p53 gene transfer system may have the potential to be applied in gynecologic malignant tumors. In cervical cancer, the use of adenoviruses as live vectors which encode wild-type p53 has given promising results. Su et al reported that recombinant adenovirus-p53 transfer combined with radiotherapy improved radiotherapeutic survival rates in patients with cervical cancer.” in Clinical application using p53 part, Line 314 and above reference (Ref 115) in n revised manuscript.
We also added “The use of adenoviruses as live vectors which encode wild-type p53 has given promising results in cervical cancer patients in abstract part, Line 26 in revised manuscript.
We deleted “but not gynecologic tumors” in abstract part, Line 23 and “Although gene therapy targeting p53 in gynecologic malignant tumors has not been developed, an adenovirus-mediated p53 gene transfer system may have the potential to be applied for gynecologic malignant tumor patients with TP53 mutations.” in Clinical application using p53 part, Line 289 in old manuscript.
We changed “other cancers” to “malignant tumors” in Clinical application using p53 part, Line 257 in old manuscript or Line 280 in revised manuscript.
[4] Conclusions:
Please, make a comment about the unmet need for the development of molecularly targeted drugs and immune system modulation for patients with persistent, metastatic and recurrent cervical cancer. Additional biomarkers may help guide therapy for patients who progress on antiangiogenesis therapy in the absence of therapeutic choices.
In a study, next-generation sequencing of 224 tumour samples identified mutational hot spots corresponding to PI3KCA (26%), BRCA2 (21%), BRCA1 (10%),KRAS (10%), TP53 (10%), and FBXW7 (10%). Gene amplification of EGFR (11%) and HER2 (8%) was also demonstrated. These results suggest that poly ADP ribose polymerase (PARP) inhibition,PI3K/AKT/mTOR pathway inhibitors, EGFR- and HER2-directed therapy, may be promising areas for future research in advanced cervical cancer.
Recommended reference:
Feldman R, et al. Paving the road to personalized medicine in cervical cancer: Theranostic biomarker evaluation in a 592-specimen library. Gynecologic oncology. 2015;137:141. Supplement 1(0).
We added “It is necessary for the development of molecularly targeted drugs and immune system modulation for patients with persistent, metastatic and recurrent cervical cancer. Additional biomarkers may help guide therapy for patients who progress on antiangiogenesis therapy in the absence of therapeutic choices. Feldmana et al demonstrated next-generation sequencing of 224 tumor samples identified mutational hot spots corresponding to PI3KCA (26%), BRCA2 (21%), BRCA1 (10%), KRAS (10%), TP53 (10%), and FBXW7 (10%). Gene amplification of EGFR (11%) and HER2 (8%) was also demonstrated. These results suggest that poly ADP ribose polymerase (PARP) inhibition, PI3K/AKT/mTOR pathway inhibitors, EGFR- and HER2-directed therapy, may be promising areas for future research in advanced cervical cancer.” in conclusions part, Line 339 and above reference (Ref 119) in revised manuscript.
Thank you again for your kindly review and comments.
Reviewer 3 Report
Review of “The association and significance of p53 in gynecologic cancers: The potential of targeted therapy”.
The authors present a well written review on the current status of p53 mutation/dysfunction in various gynecologic cancers such as endometrial, cervical, and ovarian cancer. The authors discuss mechanisms by which p53 function is altered in various gynecologic cancers, including but not limited to HPV oncoproteins E6 and E7, MDM2/X status, and p53 mutational spectrum. The literature review is up to date with respect to both mechanism by which p53 alteration induces malignant transformation as well as clinical prognostic correlates between p53 status and clinical stage/grade. The figures are well done and informative and the review contains a sufficient explanation of the newer hypothesis of ovarian oncogenesis, namely the model proposed by Christopher Crum and Raymond (Andreas) DuBois on p53 signatures and early dysplastic lesions in the fimbria and fallopian tubes leading to high-grade serous ovarian cancer.
A brief discussion of recent clinical trials using p53 gene therapy and MDM2 modulating agents such as nutlins is given.
Overall this is a good review on p53 and gynecologic cancers, which should be looked upon by the field favorably.
Some minor suggestions on the paper which could use clarification:
In the abstract on line 17 please explain what you mean by “dysregulation of the factors”, Which factors? In the Introduction on line 35 could use a general reference to all the roles of p53 like “as reviewed in “. Throughout the paper the phrase p53 alteration of loss of function causes carcinogenesis. However in many gynecologic cancers p53 mutation or dysregulation is considered a later event and in some cases not causal for initial oncogenesis. A better phrasing than “causes” might be more appropriate. In figure 3 please refer to Christopher Crum’s hypothesis on origins of ovarian cancer. Although this theory of the cells of the mullerian system and fimbria/fallopian tube as being the putative cell of origin there is still ample evidence that many ovarian cancer (including high grade serous) can arise from the ovarian surface epithelium, including a nice paper from Jeff Boyd which shows that p53 mutational spectrum of epithelial cells in dysplastic lesions like inclusion cysts and deep surface invaginations are seen in the contiguous surface epithelium as well as underlying malignant cells in the ovary. Some discussion of the alterative hypothesis should be given. There are some studies given with relation to MDM2 inhibitors and p53 gene therapy however most of the studies referenced are in leukemia. A search of clinicaltrials.gov has a number of clinical trial on p53 gene therapy, and nutlins (MDM2 inhibitors) in various gynecologic cancers. These can be mentioned here even if they are still open and/or results have not been published. there also should be some discussion of the recent literature on Rb and RBL2 with respect to p53 mutational status as there is recent findings which may impact the ability of p53 directed therapies. A brief discusion on this should be given.
Overall, this is a well written paper and recommended, with some minor revision, for publication.
Author Response
Dear reviewer 3
Thank you for your thoughtful and constructive comments. According to your suggestions, we extensively revised our manuscript as follows.
・In the abstract on line 17 please explain what you mean by “dysregulation of the factors”, Which factors?
We added “such as ERβ and MDM2” in abstract part, Line 17 and in conclusions part, Line 333 in revised manuscript, respectively.
・In the Introduction on line 35 could use a general reference to all the roles of p53 like “as reviewed in “.
We added as reference 3 in revised manuscript.
・Throughout the paper the phrase p53 alteration of loss of function causes carcinogenesis. However in many gynecologic cancers p53 mutation or dysregulation is considered a later event and in some cases not causal for initial oncogenesis. A better phrasing than “causes” might be more appropriate.
We changed the phrase “causes” to “is closely associated with” in introduction part, Line 43 and in Significance of p53 in endometrial cancer part, Line 136 in revised manuscript, respectively.
・In figure 3 please refer to Christopher Crum’s hypothesis on origins of ovarian cancer. Although this theory of the cells of the mullerian system and fimbria/fallopian tube as being the putative cell of origin there is still ample evidence that many ovarian cancer (including high grade serous) can arise from the ovarian surface epithelium, including a nice paper from Jeff Boyd which shows that p53 mutational spectrum of epithelial cells in dysplastic lesions like inclusion cysts and deep surface invaginations are seen in the contiguous surface epithelium as well as underlying malignant cells in the ovary.
We added “Figure 3 shows carcinogenesis of ovarian serous carcinoma based on crum’s hypothesis [97].” in Significance of p53 in ovarian cancer part, Line 255 and addes as Ref 97 in revised manuscript.
We discussed Jeff Boyd’s data in Significance of p53 in ovarian cancer part, Line 257 and added as Ref 98 in revised manuscript.
・Some discussion of the alterative hypothesis should be given. There are some studies given with relation to MDM2 inhibitors and p53 gene therapy however most of the studies referenced are in leukemia. A search of clinicaltrials.gov has a number of clinical trial on p53 gene therapy, and nutlins (MDM2 inhibitors) in various gynecologic cancers. These can be mentioned here even if they are still open and/or results have not been published.
We added “Clinical phase I trials using idasanutlin are going in solid tumors” in Line 309 and “In ovarian cancer, clinical phase II trials using p53 Synthetic Long Peptides Vaccine with Cyclophosphamide are going” in Line 326, respectively, and also added as Ref 133 in revised manuscript.
・there also should be some discussion of the recent literature on Rb and RBL2 with respect to p53 mutational status as there is recent findings which may impact the ability of p53 directed therapies. A brief discusion on this should be given.
We discussed expression of RBL2 in cervical and ovarian cancers in Clinical application using p53 part, Line 320 and also added as Ref 117 and 118, in revised manuscript, respectively.
Thank you again for your kindly review and comments.